# Diffusion-based Molecule Generation with Informative Prior Bridges

**Lemeng Wu**[*]
University of Texas at Austin
lmwu@cs.utexas.edu

**Chengyue Gong**[*]
University of Texas at Austin
cygong@cs.utexas.edu

**Xingchao Liu**
University of Texas at Austin
xcliu@cs.utexas.edu

**Mao Ye**
University of Texas at Austin
my21@cs.utexas.edu

**Qiang Liu**
University of Texas at Austin
lqiang@cs.utexas.edu

## Abstract

AI-based molecule generation provides a promising approach to a large area of biomedical sciences and engineering, such as antibody design, hydrolase engineering, or vaccine development. Because the molecules are governed by physical laws, a key challenge is to incorporate prior information into the training procedure to generate high-quality and realistic molecules. We propose a simple and novel approach to steer the training of diffusion-based generative models with physical and statistics prior information. This is achieved by constructing physically informed diffusion bridges, stochastic processes that guarantee to yield a given observation at the fixed terminal time. We develop a Lyapunov function based method to construct and determine bridges, and propose a number of proposals of informative prior bridges for both high-quality molecule generation and uniformity-promoted 3D point cloud generation. With comprehensive experiments, we show that our method provides a powerful approach to the 3D generation task, yielding molecule structures with better quality and stability scores and more uniformly distributed point clouds of high qualities.

## 1 Introduction

As exemplified by the success of AlphafoldV2 [22] in solving protein folding, deep learning techniques have been creating new frontiers on molecular sciences [46]. In particular, the problem of building deep generative models for molecule design has attracted increasing interest with a magnitude of applications in physics, chemistry, and drug discovery [*e.g.*, 2, 3, 26]. Recently, diffusion-based generative model have been applied to molecule generation problems [9, 19] and obtain superior performance. The idea of these methods is to corrupt the data with diffusion noise and learn a neural diffusion model to revert the corruption process to generate meaningful data from noise.

A key challenge in deep generative models for molecule and 3D point generation is to efficiently incorporate strong prior information to reflect the physical and problem-dependent statistical properties of the problems at hand. In fact, a recent fruitful line of research [11, 23, 37] have shown promising results by introducing inductive bias into the design of model architectures to reflect physical constraints such as SE(3) equivariance. In this work, we present a different paradigm of prior incorporation tailored to diffusion-based generative models, and leverage it to yield substantial improvement in both 1) high-quality and stable molecule generation and 2) uniformity-promoted point cloud generation. Our contributions are summarized as follows.

---

[*]Equal contribution

36th Conference on Neural Information Processing Systems (NeurIPS 2022).

**Prior Guided Learning of Diffusion Models.** We introduce a simple and flexible framework for injecting informative problem-dependent prior and physical information when learning diffusion-based generative models. The idea is to elicit and inject prior information regarding how the diffusion process should look like for generating each given data point, and train the neural diffusion model to imitate the prior processes. The prior information is presented in the form of diffusion bridges which are diffusion processes that are guaranteed to generate each data point at the fixed terminal time. We provide a general Lyapunov approach for constructing and determining bridges and leverage it to develop a way to systematically incorporate prior information into bridge processes.

**Physics-informed Molecule Generation.** We apply our method to molecule generation. We propose a number of energy functions for incorporating physical and statistical prior information. Compared with existing physics-informed molecule generation methods [*e.g.*, 9, 14, 29, 14], our method modifies the training process, rather than imposing constraints on the model architecture. Experiments show that our method achieves current state-of-the-art generation quality and stability on multiple test benchmarks of molecule generation.

**Uniformity-promoting Point Generation.** A challenging task in physical simulation, graphics, 3D vision is to generate point clouds for representing real objects [*e.g.*, 1, 5, 28, 51]. A largely overlooked problem of existing approaches is that they tend to generate unevenly distributed points, which lead to unrealistic shapes and make the subsequent processing and applications, such as mesh generation, challenging and inefficient. In this work, we leverage our framework to introduce uniformity-promoting forces into the prior bridge of diffusion generative models. This yields a simple and efficient approach to generating regular and realistic point clouds in terms of both shape and point distribution.

## 2 Related works

**Diffuse Bridge Process.** Diffusion-based generative models [18, 40, 41, 44, 25] have achieved great successes in various AI generation tasks recently; these methods leverage a time reversion technique and can be viewed as learning variants auto-encoders with diffusion processes as encoders and decoders. Schrodinger bridges [7, 9, 45] have also been proposed for learning diffusion generative models that guarantee to output desirable outputs in a finite time interval, but these methods involve iterative proportional fittings and are computationally costly. Our framework of learning generative models with diffusion bridges is similar to that of [35], which learn diffusion models as a mixture of forward-time diffusion bridges to avoid the time-reversal technique of [43]. But our framework is designed to incorporate physical prior into bridges and develop a systematic approach for constructing a broad class of prior-informed bridges.

**3D Molecule Generation.** Generating molecule in 3D space has been gaining increasing interest. A line of works [*e.g.* 30, 32, 38, 39, 48, 49, 50] consider conditional conformal generation, which takes the 2D SMILE structure as conditional input and generate the 3D molecule conformations condition on the input. Another series of works [*e.g.*, 13, 19, 27, 37, 47] focus on directly generating the atom position and type for the molecule unconditionally. For these series of works, improvements usually come from architecture design and loss design. For example, G-Schnet [13] auto-regressively generates the atom position and type one by one after another; EN-Flow [37] and EDM [19] adopt E(n) equivariant graph neural network (EGNN) [37] to train flow-based model and diffusion model. These methods aim at generating valid and natural molecules in 3D space and outperform previous approaches by a large margin. Our work provides a very different approach to incorporating the physical information for molecule generation by injecting the prior information into the diffusion process, rather than neural network architectures.

**Point Cloud Generation.** A vast literature has been devoted to learning deep generative models for real-world 3D objects in the form of point clouds. [1] first proposed to generate the point cloud by generating a latent code and training a decoder to generate point clouds from the latent code. Build upon this approach, methods have been developed using flow-based generative models [51] and diffusion-base models [5, 28, 29]. However, the existing works miss a key important prior information: the points in a point cloud tend to distribute regularly and uniformly. Ignoring this information causes poor generation quality. By introducing uniformity-promoting forces in diffusion bridges, we obtain a simple and efficient approach to generating regular and realistic point clouds.

# 3  Method

We first introduce the definition of diffusion generative models and discuss how to learn these models with prior bridges. After introducing the training algorithm for deep diffusion generative models, we discuss the energy functions that we apply to molecules and point cloud examples.

## 3.1  Learning Diffusion Generative Models with Prior Bridges

**Problem Definition.** We aim at learning a generative model given a dataset $\{x^{(k)}\}_{k=1}^{n}$ drawn from an unknown distribution $\Pi^*$ on $\mathbb{R}^d$. A diffusion model on time interval $[0, 1]$ is

$$\mathbb{P}^\theta: \quad dZ_t = s_t^\theta(Z_t)dt + \sigma_t(Z_t)dW_t, \quad \forall t \in [0, 1], \quad Z_0 \sim \mu_0,$$

where $W_t$ is a standard Brownian motion; $\sigma_t \colon \mathbb{R}^d \to \mathbb{R}^{d \times d}$ is a positive definition covariance coefficient; $s_t^\theta \colon \mathbb{R}^d \to \mathbb{R}^d$ is parameterized as a neural network with parameter $\theta$, and $\mu_0$ is the initialization. Here we use $\mathbb{P}^\theta$ to denote the distribution of the whole trajectory $Z = \{Z_t \colon t \in [0, 1]\}$, and $\mathbb{P}_t^\theta$ the marginal distribution of $Z_t$ at time $t$. We want to learn the parameter $\theta$ such that the distribution $\mathbb{P}_1^\theta$ of the terminal state $Z_1$ equals the data distribution $\Pi^*$.

**Learning Diffusion Models.** There are an infinite number of diffusion processes $\mathbb{P}^\theta$ that yield the same terminal distribution but have different distributions of latent trajectories $Z$. Hence, it is important to inject problem-dependent prior information into the learning procedure to obtain a model $\mathbb{P}^\theta$ that simulate the data for the problem at hand fast and accurately. To achieve this, we elicit an *imputation* process $\mathbb{Q}^x$ for each $x \in \mathbb{R}^d$, such that a draw $Z \sim \mathbb{Q}^x$ yields trajectories that 1) are consistent with $x$ in that $Z_1 = x$ deterministically, and 2) reflect important physical and statistical prior information on the problem at hand.

Formally, if $\mathbb{Q}^x(Z_1 = x) = 1$, we call that $\mathbb{Q}^x$ is a bridge process pinned at end point $x$, or simply an $x$-bridge. Assume we first generate a data point $x \sim \Pi^*$, and then draw a bridge $Z \sim \mathbb{Q}^x$ pinned at $x$, then the distribution of $Z$ is a mixture of $\mathbb{Q}^x$ with $x$ drawn from the data distribution: $\mathbb{Q}^{\Pi^*} := \int \mathbb{Q}^x(\cdot)\Pi^*(dx)$.

A key property of $\mathbb{Q}^{\Pi^*}$ is that its terminal distribution equals the data distribution, i.e., $\mathbb{Q}_1^{\Pi^*} = \Pi^*$. Therefore, we can learn the diffusion model $\mathbb{P}^\theta$ by fitting the trajectories drawn from $\mathbb{Q}^{\Pi^*}$ with the "backward" procedure above. This can be formulated by maximum likelihood or equivalently minimizing the KL divergence:

$$\min_\theta \left\{ \mathcal{L}(\theta) := \mathcal{KL}(\mathbb{Q}^{\Pi^*} \mid\mid \mathbb{P}^\theta) \right\}.$$

Furthermore, assume that the bridge $\mathbb{Q}^x$ is a diffusion model of form

$$\mathbb{Q}^x: \quad dZ_t = b_t(Z_t \mid x)dt + \sigma_t(Z_t)dW_t, \quad Z_0 \sim \mu_0, \tag{1}$$

where $b_t(Z_t \mid x)$ is an $x$-dependent drift term need to carefully designed to both satisfy the bridge condition and incorporate important prior information (see Section 3.2). Assuming this is done, using Girsanov theorem [33], the loss function $\mathcal{L}(\theta)$ can be reformed into a form ofdenoised score matching loss of [*e.g.*, 41, 43, 42]:

$$\mathcal{L}(\theta) = \mathbb{E}_{Z \sim \mathbb{Q}^{\Pi^*}} \left[ \frac{1}{2} \int_0^1 \left\| \sigma(Z_t)^{-1}(s_t^\theta(Z_t) - b_t(Z_t \mid Z_1)) \right\|_2^2 dt \right] + \text{const}, \tag{2}$$

which is a score matching term between $s^\theta$ and $b$. The const term contains the log-likelihood for the initial distribution $\mu_0$, which is a const in our problem. Here $\theta^*$ is an global optimum of $\mathcal{L}(\theta)$ if

$$s_t^{\theta^*}(z) = \mathbb{E}_{Z \sim \mathbb{Q}^{\Pi^*}}[b_t(z|Z_1) \mid Z_t = z].$$

This means that the drift term $s_t^\theta$ should be matched with the conditional expectation of $b_t(z|x)$ with $x = Z_1$ conditioned on $Z_t = z$.

**Remark 3.1.** *The SMLD can be viewed as a special case of this framework when we take $\mathbb{Q}^x$ to be a time-scaled Brownian bridge process:*

$$\mathbb{Q}^{x,\text{bb}}: \qquad dZ_t = \sigma_t^2 \frac{x - Z_t}{\beta_1 - \beta_t} dt + \sigma_t dW_t, \quad Z_0 \sim \mathcal{N}(x, \beta_1), \tag{3}$$

where $\sigma_t \in [0, +\infty)$ and $\beta_t = \int_0^t \sigma_s^2 \mathrm{d}s$. This can be seen by the fact that the time-reversed process $\tilde{Z}_t := Z_{1-t}$ follows the simple time-scaled Brownian motion $\mathrm{d}\tilde{Z}_t = \sigma_{1-t}\mathrm{d}\tilde{W}_t$ starting from the data point $\tilde{Z}_0 = x$, where $\tilde{W}_t$ is another standard Brownian motion. The Brownian bridge achieves $Z_1 = x$ because the magnitude of the drift force is increasing to infinite when $t$ is close to time $1$.

However, the bridge of SMLD above is a relative simple and uninformative process and does not incorporate problem-dependent prior information into the learning procedure. This is also the case of the other standard diffusion-based models [43], such as denoising diffusion probabilistic models (DDPM) which can be shown to use a bridge constructed from an Ornstein–Uhlenbeck process. We refer the readers to [35], which provides a similar forward time bridge framework for learning diffusion models, and it recovers the bridges in SMLD and DDPM as a conditioned stochastic process derived using the $h$-transform technique [10]. However, the $h$-transform method is limited to elementary stochastic processes that have an explicit formula of the transition probabilities, and can not incorporate complex physical statistical prior information. Our work strikes to construct and use a broader class of more complex bridge processes that both reflect problem-dependent prior knowledge and satisfy the endpoint condition $\mathbb{Q}^x(Z_1 = x) = 1$. This necessitate systematic techniques for constructing a large family of bridges, as we pursuit in Section 3.2.

## 3.2 Designing Informative Prior Bridges

The key to realizing the general prior-informed learning framework above is to have a general and user-friendly technique to design $\mathbb{Q}^x$ in (1) to ensure the bridge condition $\mathbb{Q}^x(Z_1 = x) = 1$ while leaving the flexibility of incorporating rich prior information. To achieve this, we first develop a general criterion of bridges based on a *Lyapunov function method* which allows us to identify a very general form of bridge processes; we then propose a particularly simple family of bridges that we use in practice by introducing modification to Brownian bridges.

**Definition 3.2 (Lyapunov Functions).** *A function $U_t(z)$ is said to be a Lyapunov function for set $A \subset \mathbb{R}^d$ at time $t = 1$ if $U_1(z) \geq 0$ for $\forall z \in \mathbb{R}^d$ and $U_1(z) = 0$ if and only if $z \in A$.*

Intuitively, a diffusion process $\mathbb{Q}$ is a bridge $A$, i.e., $\mathbb{Q}(Z_1 \in A) = 1$, if it (at least) approximately follows the gradient flow of a Lyapunov function and the magnitude (or step size) or the gradient flow should increase with a proper magnitude in order to ensure that $Z_t \in A$ at the terminal time $t = 1$. Therefore, we identify a general form of bridges to $A$ as follows:

$$\mathbb{Q}^A: \quad \mathrm{d}Z_t = (-\alpha_t \nabla_z U_t(Z_t) + \nu_t(Z_t))\,\mathrm{d}t + \sigma_t(Z_t)\mathrm{d}W_t, \quad t \in [0,1], \quad Z_0 \sim \mu_0, \quad (4)$$

where $\alpha_t > 0$ is the step size of the gradient flow of $U$ and $\nu$ is an extra perturbation term. The step size $\alpha_t$ should increase to infinity as $t \to 1$ sufficiently fast to dominate the effect of the diffusion term $\sigma_t \mathrm{d}W_t$ and the perturbation $\nu_t \mathrm{d}t$ term to ensure that $U$ is minimized at time $t = 1$.

**Proposition 3.3.** *Assume $U_t(z) = U(z, t)$ is a Lyapunov function of a measurable set $A$ at time $1$ and $U(\cdot, t) \in C^2(\mathbb{R}^d)$ and $U(z, \cdot) \in C^1([0, 1])$. Then, $\mathbb{Q}^A$ in (4) is an bridge to A, i.e., $\mathbb{Q}^A(Z_1 \in A) = 1$, if the following holds:*

*1) $U$ follows an (expected) Polyak-Lojasiewicz condition: $\mathbb{E}_{\mathbb{Q}^A}[U_t(Z_t)] - \|\nabla_z U_t(Z_t)\|^2 \leq 0, \forall t.$*

*2) Let $\beta_t = \mathbb{E}_{\mathbb{Q}^A}[\nabla_z U_t(Z_t)^\top \nu_t(Z_t)]$, and $\gamma_t = \mathbb{E}_{\mathbb{Q}^A}[\partial_t U_t(Z_t) + \frac{1}{2}\mathrm{tr}(\nabla_z^2 U_t(Z_t)\sigma_t^2(Z_t))]$, and $\zeta_t = \exp(\int_0^t \alpha_s \mathrm{d}s)$. Then $\lim_{t\uparrow 1} \zeta_t = +\infty$, and $\lim_{t\uparrow 1} \frac{\zeta_t}{\int_0^t \zeta_s(\beta_s + \gamma_s)\mathrm{d}s} = +\infty.$*

Brownian bridge can be viewed as the case when $U_t(z) = \|x - z\|^2/2$ and $\alpha_t = \sigma_t^2/(\beta_1 - \beta_t)$, and $\nu = 0$. Hence simply introducing an extra drift term into bridge bridge yields that a broad family of bridges to $x$:

$$\mathbb{Q}^{x,\mathrm{bb},f}: \quad \mathrm{d}Z_t = \left(\sigma_t f_t(Z_t) + \sigma_t^2 \frac{x - Z_t}{\beta_1 - \beta_t}\right)\mathrm{d}t + \sigma_t \mathrm{d}W_t, \quad Z_0 \sim \mu_0. \quad (5)$$

In Appendix A.4 and A.5, we show that $\mathbb{Q}^{x,\mathrm{bb},f}$ is a bridge to $x$ if $\mathbb{E}_{\mathbb{Q}^{x,\mathrm{bb}}}[\|f_t(Z_t)\|^2] < +\infty$ and $\sigma_t > 0, \forall t$, which is very mild condition and is satisfied for most practical functions. The intuition is that the Brownian drift $\sigma_t^2 \frac{x - Z_t}{\beta_1 - \beta_t}$ is singular and grows to infinite as $t$ approaches 1. Hence,

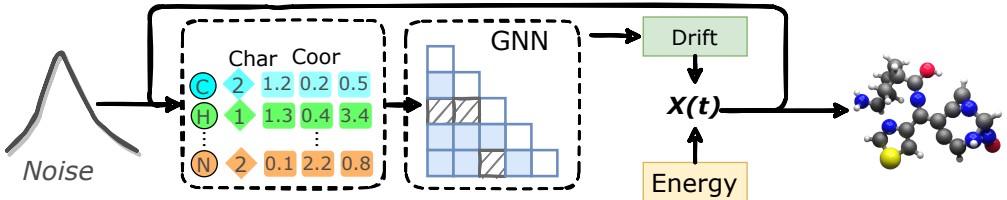

Figure 1: An overview of our training pipeline with molecule generation as an example. Initialized from a given distribution, we pass the data through the network multiple times, and finally get the meaningful output.

introducing an $f$ into the drift would not change of the final bridge condition, unless $f$ is also singular and has a magnitude that dominates the Brownian bridge drift as $t \to 1$.

To make the model $\mathbb{P}^\theta$ compatible with the physical force $f$, we assume the learnable drift has a form of $s_t^\theta(z) = \alpha f_t(z) + \tilde{s}_t^\theta(z)$ where $\tilde{s}$ is a neural network (typically a GNN) and $\alpha$ can be another learnable parameter or a pre-defined parameter. Please refer to algorithm 3.2 and Figure 1 for descriptions about our practical algorithm.

---

**Algorithm 1** Learning diffusion generative models.

---

**Input**: Given a dataset $\{x^{(k)}\}$, $\mathbb{Q}^x$ the bridge in (5), and a problem-dependent prior force $f$,v and a diffusion model $\mathbb{P}^\theta$.
**Training**: Estimate $\theta$ by minimizing $\mathcal{L}(\theta)$ in (2) with stochastic gradient descent and time discretization.
**Sampling**: Simulate from $\mathbb{P}^\theta$.

---

## 4 Molecule and 3D Generation with Informative Prior Bridges

We apply our method to the molecule generation as well as point cloud generation. Informative physical or statistical priors that reflects the underlying real physical structures can be particularly beneficial for molecule generation as we show in experiments.

In our problem, each data point $x$ is a collection of atoms of different types, more generally marked points, in 3D Euclidean space. In particular, we have $x = [x_i^r, x_i^h]_{i=1}^m$, where $x_i^r \in \mathbb{R}^3$ is the coordinate of the $i$-th atom, and $x_i^h \in \{e_1, \ldots, e_k\}$ where each $e_i = [0 \cdots 1 \cdots 0]$ is the $i$-th basis vector of $\mathbb{R}^k$, which indicates the type of the $i$-th atom of $k$ categories. To apply the diffusion generative model, we treat $x_i^h$ as a continuous vector in $\mathbb{R}^r$ and round it to the closest basis vector when we want to output a final result or have computations that depend on atom types (e.g., calculating an energy function as we do in sequel). Specifically, for a continuous $x_i^h \in \mathbb{R}^k$, we denote by $\hat{x}_i^h = \mathbb{I}(x_i^h = \max(x_i^h))$ the discrete type rounded from it by taking the type with the maximum value. To incorporate priors, we design an energy function $E(x)$ and incorporate $f_t(\cdot) = -\nabla E(\cdot)$ into the Brownian bridge (5) to guide the training process. We discuss different choices of $E$ in the following.

### 4.1 Prior Bridges for Molecule Generation

Previous prior guided molecule or protein 3D structure generation usually depends on pre-defined energy or force [30, 50]. We introduce our two potential energies. One is formulated inspired by previous works in biology, and the other is an $k$ nearest neighbour statistics directly obtained from the data.

**AMBER Inspired Physical Energy.** AMBER [12] is a family of force fields for molecule simulation. It is designed to provide a computationally efficient tool for modern chemistry-molecular dynamics and free energy calculations. It consists of a number of important forces, including the bond energy, angular energy, torsional energy, the van der Waals energy and the Coulomb energy. Inspired by AMBER, we propose to incorporate the following energy term into the bridge process:

$$E(x) = E_{bond}(x) + E_{angle}(x) + E_{LJ}(x) + E_{Coulomb}(x). \tag{6}$$

- The bond energy is $E_{bond}(x) = \sum_{ij \in bond(x)}(\text{Len}(x^r_{ij}) - \ell(\hat{x}^h_i, \hat{x}^h_j))^2$, where $\text{Len}(x^r_{ij}) = \left\| x^r_i - x^r_j \right\|$, and $bond(x)$ denotes the set of bonds from $x$, which is set to be the set of atom pairs with a distance smaller than 1.15 times the covalent radius; the $\ell^0(r, c)$ denotes the expected bond length between atom type $r$ and $c$, which we calculate as side information from the training data.

- The angle energy is $E_{angle}(x) = \sum_{ijk \in angle(x)}(\text{Ang}(x^r_{ijk}) - \omega^0(\hat{x}^h_{ijk}))^2$, where $angle(x)$ denotes the set of angles between two neighbour bonds in $bound(x)$, and $\text{Ang}(x^r_{ijk})$ denotes the angle formed by vector $x^r_i - x^r_j$ and $x^r_k - x^r_j$, and $\omega^0(\hat{x}^h_{ijk})$ is the expected angle between atoms of type $\hat{x}^h_i$, $\hat{x}^h_j$, $\hat{x}^h_k$, which we calculate as side information from the training data.

- The Lennard-Jones (LJ) energy is defined by $E_{LJ}(x) = \sum_{i \neq j} e(\left\| x^r_i - x^r_j \right\|)$ and $e(\ell) = (\sigma/\ell)^{12} - 2(\sigma/\ell)^6$. The parameter $\sigma$ is an approximation for average nucleus distance.

- The nuclei-nuclei repulsion (Coulomb) electromagnetic potential energy is $E_{Coulomb}(x) = \kappa \sum_{ij} q(\hat{x}^h_i) q(\hat{x}^h_j) / \left\| x^r_i - x^r_j \right\|$, where $\kappa$ is Coulomb constant and $q(r)$ denotes the point charge of atom of type $r$, which depends on the number of protons.

**Statistical Energy.** When accurate physic laws are unavailable, molecular geometric statistics, such as bond lengths, bond angles, and torsional angles, etc, can be directly calculated from the data and shed important insights on the system [*e.g.*, 8, 21, 31]. We propose to design a prior energy function in bridges by directly calculate these statistics over the dataset.

Specifically, we assume that the lengths and angles of each type of bond follows a Gaussian distribution that we learn from the dataset, and define the energy function as the negative log-likelihood:

$$E_{stat}(x) = \sum_{ij \in knn(x)} \frac{1}{\hat{\sigma}^2_{\hat{x}^h_{ij}}} \left\| \text{Len}(x^r_{ij}) - \hat{\mu}_{\hat{x}^h_{ij}} \right\|^2 + \sum_{ij,jk \in knn(x)} \frac{1}{\sigma^2_{\hat{x}^h_{ijk}}} \left\| \text{Ang}(x^r_{ijk}) - \mu_{\hat{x}^h_{ijk}} \right\|^2, \quad (7)$$

where $knn(x)$ denotes the K-nearest neighborhood graph constructed based on the distance matrix of $x$; for each pair of atom types $r, c \in [k]$, $\hat{\mu}_{rc}$ and $\hat{\sigma}^2_{rc}$ denotes empirical mean and variance of length of $rc$-edges in the dataset; for each triplet $r, c, r' \in [k]$, $\hat{\mu}_{rcr'}$ and $\hat{\sigma}^2_{rcr'}$ is the empirical mean and variance of angle betwen $rc$ and $cr'$ bonds.

Intuitively, depending on the atom type and order of the nearest neighbour, we force the atom distance and angle to mimic the statistics calculated from the data. We thus implicitly capture different kinds of interaction forces. Compared with the AMBER energy, the statistical energy (7) is simpler and more adaptive to the dataset of interest.

## 4.2 Prior Bridges for Point Cloud Generation

We design prior forces for 3D point cloud generation, which is similar to molecule generation except that the points are un-typed so we only have the coordinates $\{x^r_i\}$. One important aspect of point cloud generation is to distribute points uniformly on the surface, which is important for producing high-quality meshes and other post-hoc geometry applications and manipulations.

**Riesz Energy.** One idea to make the point distribute uniformly is adding a repulsive force to separate the points apart from each other [24, 15, 16]. We achieve this by minimizing the Riesz energy [17],

$$E_{\text{Riesz}}(x) = \frac{1}{2} \sum_{j \neq i} ||x^r_i - x^r_j||^{-2}. \quad (8)$$

**KNN Distance Energy.** Similar to molecule design, we directly calculate the average distance between each point and its k nearest neighbour neighbour, and define the following energy:

$$E_{\text{knn}}(x) = \sum_i \left( \text{knn-dist}_i(x^r) - \mu_{knn} \right)^2, \quad (9)$$

where $\text{knn-dist}_i(x) = \frac{1}{K} \sum_{j \in \mathcal{N}_K(x_i;x)} ||x^r_i - x^r_j||^2$ denotes the average distance from $x_i$ to its $K$ nearest neighbors, and $\mu_{knn}$ is the empirical mean of $\text{knn-dist}_i(x)$ in the dataset. This would encourage the points to have similar average nearest neighbor distance and yield uniform distribution between points. In common geometric setups, the valence of the point on the surface is 4, which means we set $k = 4$.

## 5 Experiment

We verify the advantages of our proposed method (Bridge with Priors) in several different domains. We first compare our method with advanced generators (*e.g.*, diffusion model, normalizing flow, etc.) on molecule generation tasks. We then implement our method on point cloud generations, which targets producing generated samples in a higher quality. We directly compare the performance and also analyze the difference between our energy prior and other energies we discuss in Section 3.

### 5.1 Force Guided Molecule Generation

To demonstrate the efficiency and effectiveness of our bridge processes and physical energy, we conduct experiments on molecule and macro-molecule generation experiments. We follow [29] in settings and observe that our proposed prior bridge processes consistently improve the state-of-the-art performance. Diving deeper, we analyze the impact of different energy terms and hyperparameters.

**Metrics.** Following [19, 37], we use the atom and molecular stability score to measure the model performance. The atom stability is the proportion of atoms that have the right valency while the molecular stability stands for the proportion of generated molecules for which all atoms are stable. For visualization, we use the distance between pairs of atoms and the atom types to predict bond types, which is a common practice. we extracted 10,000 samples to calculate the above metrics.

**Dataset Settings** QM9 [36] molecular properties and atom coordinates for 130k small molecules with up to 9 heavy atoms with 5 different types of atoms. This data set contains small amino acids, such as GLY, ALA, as well as nucleobases cytosine, uracil, and thymine. We follow the common practice in [19] to split the train, validation, and test partitions, with 100K, 18K, and 13K samples. GEOM-DRUG [4] is a dataset that contains drug-like molecules. It features 37 million molecular conformations annotated by energy and statistical weight for over 450,000 molecules. Each molecule contains 44 atoms on average, with 5 different types of atoms. Following [19, 37], we retain the 30 lowest energy conformations for each molecule.

**Training Configurations.** On QM9, we train the EGNNs with 256 hidden features and 9 layers for 1100 epochs, a batch size 64, and a constant learning rate $10^{-4}$, which is the default training configuration. We use the polynomial noise schedule used in [19] which linearly decay from $10^{-2}/T$ to 0. We linearly decay $\alpha$ from $10^{-3}/T$ to 0 *w.r.t.* time step. We set $k = 5$ (7) by default. On GEOM-DRUG, we train the EGNNs with 256 hidden features and 8 layers with batch size 64, a constant learning rate $10^{-4}$, and 10 epochs. It takes approximately 10 days to train the model on these two datasets on one `Tesla V100-SXM2-32GB` GPU. We provide E(3) Equivariant Diffusion Model (EDM) [19] and E(3) Equivariant Normalizing Flow (EN-Flow) [37] as our baselines. Both two are trained with the same configurations as ours.

Table 1: Results of our method and several baselines on QM9 and GEOM-DRUG. We evaluate the percentage of valid and unique molecules out of 12000 generated molecules.

|  | QM9 | | | GEOM-DRUG | |
|---|---|---|---|---|---|
|  | Atom Sta (%) ↑ | Mol Sta (%) ↑ | Valid + Unique ↑ | Atom Sta (%) ↑ | Mol Sta (%) ↑ |
| EN-Flow [37] | 85.0 | 4.9 | 0.349 | 75.0 | 0.0 |
| GDM [19] | 97.0 | 63.2 | - | 75.0 | 0.0 |
| E-GDM [19] | **98.7±0.1** | 82.0±0.4 | 0.902 | 81.3 | 0.0 |
| Bridge | **98.7±0.1** | 81.8±0.2 | 0.902 | 81.0±0.7 | 0.0 |
| Bridge + Force (7) | **98.8±0.1** | **84.6±0.3** | 0.907 | **82.4±0.8** | 0.0 |

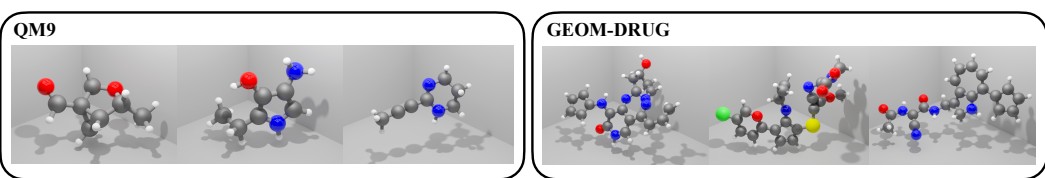

Figure 2: Examples of molecules generated by our method on QM9 and GEOM-DRUG.

**Results.** We summarize our experimental results in Table 1. We observe that **(1)** our method generates molecules with better qualities than the others. On QM9, we notice that we improve the molecule stability score by a large margin (from 82.0 to 84.6) and slightly improve the atom stability score (from 98.7 to 98.8). It indicates that with the informed prior bridge helps improves the quality of the

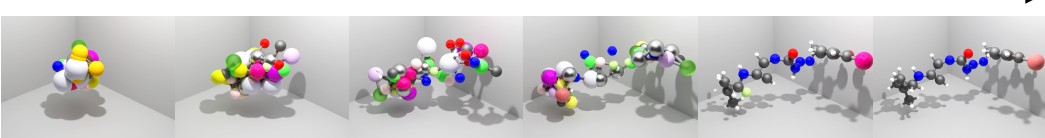

Figure 3: An example of generation trajectory following $\mathbb{P}^\theta$ of our method, trained on GEOM-DRUG.

Table 2: We compare w. and w/o force results with different discretization time steps.

| | Time Step | | | | | |
|---|---|---|---|---|---|---|
| | 50 | | 100 | | 500 | |
| | Atom Stable (%) | Mol Stable (%) | Atom Stable (%) | Mol Stable (%) | Atom Stable (%) | Mol Stable (%) |
| EGM | 97.0±0.1 | 66.4±0.2 | 97.3±0.1 | 69.8±0.2 | **98.5±0.1** | **81.2±0.1** |
| Bridge + Force (7) | **97.3±0.1** | **69.2±0.2** | **97.9±0.1** | **72.3±0.2** | **98.7±0.1** | **83.7±0.1** |

generated molecules. **(2)** On the GEOM-DRUG dataset, the atom stability is improved from 81.3 to 82.4, which shows that our method can work for macro-molecules. **(3)** We visualize and qualitatively evaluate our generate molecules. Figure 3 displays the trajectory on GEOM-DRUG and Figure 2 shows the samples on two datasets. **(4)** Bridge processes and E-GDM obtain comparable results on our tested benchmarks. **(5)** The computational load added by introducing prior bridges is small. Compared to EGM, we only introduce 8% additional cost in training and 3% for inference.

**Result: Better With Fewer Time Steps.** We display the performance of our method with fewer time steps in Table 2. We observe that **(1)** with fewer time steps, the baseline EGM method gets worse results than 1000 steps in Table 1. **(2)** with 500 steps, our method still keeps a consistently good performance. **(3)** with even fewer 50 or 100 steps, our method yields a worse result than 1000 steps in Table 1, but still outperforms the baseline method by a large margin.

Table 3: We compare EGM models trained with different force mentioned in Section 3.

| Method | Atom Stable (%) | Mol Stable (%) | Method | Atom Stable (%) | Mol Stable (%) |
|---|---|---|---|---|---|
| Force (7), $k = 7$ | **98.8±0.1** | **84.5±0.2** | Force (6) | 98.7±0.1 | 83.1±0.2 |
| Force (7), $k = 5$ | **98.8±0.1** | **84.6±0.3** | Force (6) w/o. bond | 98.7±0.1 | 82.5±0.1 |
| Force (7), $k = 3$ | **98.8±0.1** | 83.9±0.3 | Force (6) w/o. angle | 98.7±0.1 | 82.4±0.2 |
| Force (7), $k = 1$ | **98.8±0.1** | 82.7±0.3 | Force (6) w/o. Long-range | 98.7±0.1 | 82.7±0.2 |

**Ablation: Impacts of Different Energies.** We apply several energies we discuss in Section 3, and compare them on the QM9 dataset. **(1)** We notice that our energy (7) gets better performance with larger $k$ when $k \leq 5$. $k = 7$ achieves comparable performance as $k = 5$. Larger $k$ also requires more computation time, which yields a trade-off between performance and efficiency. **(2)** For (6), once removing a typical term, the performance drops. **(3)** In all the cases, applying additional forces outperforms the bridge processes baseline w/o. force.

## 5.2 Force Guided Point Cloud Generation

We apply uniformity-promoting priors to point cloud generation. We apply our method based on the diffusion model for point cloud generation introduced by point cloud diffusion model [28] and compare it with the original diffusion model as well as the case of bridge processes w/o. force prior. We observe that our method yields better results in various evaluation metrics under different setups.

**Dataset.** We use the ShapeNet [6] dataset for point cloud generation. ShapeNet contains 55 categories. We select Airplane and Chair, which are the two most common categories to evaluate in recent point cloud generation works [5, 28, 51, 52]. We construct the point clouds following the setup in [28], split the train, valid and test dataset in 80%, 15% and 5% and samples 2048 points uniformly on the mesh surface.

**Evaluation Metric.** We evaluate the generated shape quality in two aspects following the previous works, including the minimum matching distance (MMD) and coverage score (COV). These scores are the two most common practices in the previous works. We use Chamfer Distance (CD) and Earth Mover's Distance (EMD) as the distance metric to compute the MMD and COV.

**Experiment Setup.** We train the model with two different configurations. The first one uses exactly the same experiment setup configuration introduced in [28]. Thus, we use the same model architecture and train the model in 100 diffuse steps with a learning rate $2 \times 10^{-3}$, batch size 128, and linear noise schedule from 0.02 to $10^{-4}$. We initial $\alpha$ with 0.1 and jointly learn it with the network. For the second setup, to evaluate the better converge speed of our method, we decrease the diffuse step from 100 to 10 with other settings the same. For the diffusion model baseline, we reproduce the number by directly using the pre-trained model checkpoint and testing it on the test set provided by the official codebase.

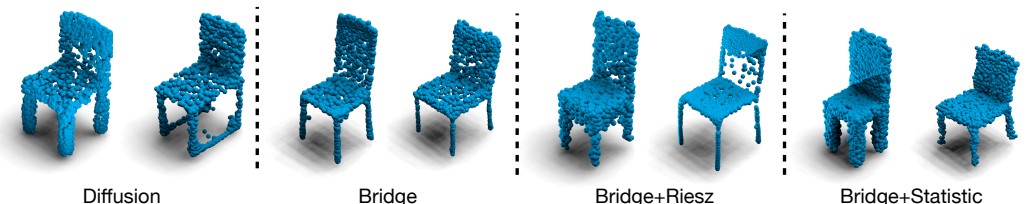

Diffusion  Bridge  Bridge+Riesz  Bridge+Statistic

Figure 4: From left to right are examples of point clouds generated by [28], our method with uninformative bridge ($f_t = 0$), bridge with Riesz energy ($f_t = -\nabla E_{\mathrm{Riesz}}$) and with KNN energy ($f_t = -\nabla E_{knn}$). We see that the Riesz and KNN energies yield more uniformly distributed points. Riesz energy sometimes creates additional outlier points due to its repulsive nature.

**Result.** We show our experimental result in Table 4. We see that **(1)** In the 10 steps setup, all variants of our approach are clearly stronger than the diffusion model. With force added, our method with physical prior achieves nearly the same performance as the 100 steps setup. **(2)** In the 100 steps setup, adding energy potential as prior improves the bridge process performance and further let it beat the diffusion model baseline.**(3)** Since the test points are uniformly sampled on the surface, a better score indicates a closer point distribution to the reference set. Further, when compare with Riesz energy (8), statistic gap energy (9) performs better. One explanation is the Riesz energy pushes the points to some outlier position in sample generate samples, while statistic gap energy is more robust. We also show visualization samples in Figure 4.

Table 4: Point cloud generation results. CD is multiplied by $10^3$, EMD is multiplied by 10.

|  |  | 10 Steps | | | | 100 Steps | | | |
|---|---|---|---|---|---|---|---|---|---|
|  |  | MMD ↓ | | COV ↑ | | MMD ↓ | | COV ↑ | |
|  |  | CD | EMD | CD | EMD | CD | EMD | CD | EMD |
| Chair | Diffusion [28] | 14.01 | 3.23 | 32.72 | 29.36 | 12.32 | 1.79 | 47.41 | **47.59** |
|  | Bridge | 13.04 | 2.14 | 46.01 | 42.59 | 12.47 | 1.85 | 47.83 | 47.13 |
|  | + Riesz | 12.84 | 1.95 | 47.21 | 44.31 | 12.31 | 1.82 | 48.14 | 47.42 |
|  | + Statistic | **12.65** | **1.84** | **47.58** | **45.23** | **12.25** | **1.78** | **48.39** | 47.56 |
| Airplane | Diffusion [28] | 3.71 | 1.31 | 43.12 | 39.94 | 3.28 | **1.04** | **48.74** | 46.38 |
|  | Bridge | 3.44 | 1.24 | 46.90 | 43.46 | 3.37 | 1.08 | 47.11 | 46.17 |
|  | + Riesz | 3.39 | 1.20 | **47.11** | 43.12 | **3.24** | 1.09 | 48.62 | 46.23 |
|  | + Statistic | **3.30** | **1.12** | 47.02 | **44.67** | **3.24** | 1.06 | 48.53 | 46.73 |

## 6    Conclusion and Limitations

We propose a framework to inject informative priors into learning neural parameterized diffusion models, with applications to both molecules and 3D point cloud generation. Empirically, we demonstrate that our method has the advantages such as better generation quality, less sampling time and easy-to-calculate potential energies. For future works, we plan to 1) study the relation between different types of forces for different domain of molecules, 2) study how to generate valid proteins in which the number of atoms is very large, and 3) apply our method to more realistic applications such as antibody design or hydrolase engineering.

In both energy functions in (7) and (6), we do not add torsional angle related energy [20] mainly because it is hard to verify whether four atoms are bonded together during the stochastic process. We plan to study how to include this for better performance in future works.

Another weakness of deep diffusion bridge processes are their computation time. Similar to previous diffusion models [29], it takes a long time to train a model. We attempted to speed the training up by

using a large batch size (*e.g.*, 512, 1024) but found a performance drop. An important future direction is to study methods to distribute and accelerate the training.

**Acknowledgements**    Authors are supported in part by CAREER-1846421, SenSE-2037267, EAGER-2041327, and Office of Navy Research, and NSF AI Institute for Foundations of Machine Learning (IFML). We would like to thank the anonymous reviewers and the area chair for their thoughtful comments and efforts towards improving our manuscript.

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
