499 # A Proofs

500 *Proof of [Proposition 3.3](#).* It is a direct result of Theorem [A.1](#). □

**Theorem A.1.** *Assume*

$$\mathrm{d}Z_t = \eta(Z_t, t)\mathrm{d}t + \sigma(Z_t, t)\mathrm{d}W_t, \qquad t \in [0, 1].$$

501 *We have $Z_1 \in A$ with probability one if there exists a function $U: \mathbb{R}^d \times [0, 1] \to \mathbb{R}$ such that*

502 *1) $U(\cdot, t) \in C^2(\mathbb{R}^d)$ and $U(z, \cdot) \in C^1([0, 1])$;*

503 *2) $U(z, 1) \geq 0$, $z \in \mathbb{R}^d$; $U(z, 1) = 0$ implies that $z \in A$, where $A$ is a measurable set in $\mathbb{R}^d$;*

504 *4) There exists a sequence $\{\alpha_t, \beta_t, \gamma_t : t \in [0, 1]\}$, such that for $t \in [0, 1]$,*

$$\mathbb{E}[\nabla_z U(Z_t, t)^\top \eta(Z_t, t)] \leq -\alpha_t \mathbb{E}[U(Z_t, t)] + \beta_t,$$

$$\mathbb{E}[\partial_t U(Z_t, t) + \frac{1}{2}\mathrm{tr}(\nabla_z^2 U(Z_t, t)\sigma^2(Z_t, t))] \leq \gamma_t;$$

505 *5) Define $\zeta_t = \exp(\int_0^t \alpha_s \mathrm{d}s)$. We assume*

$$\lim_{t \uparrow T} \zeta_t = +\infty, \quad \lim_{t \uparrow T} \frac{\zeta_t}{\int_0^t \zeta_s(\beta_s + \gamma_s)\mathrm{d}s} = +\infty. \tag{10}$$

506 *Proof.* Following $\mathrm{d}Z_t = \eta(Z_t, t)\mathrm{d}t + \sigma(Z_t, t)\mathrm{d}W_t$, we have by Ito's Lemma,

$$\mathrm{d}U(Z_t, t) = \nabla U(Z_t, t)^\top(\eta(Z_t, t)\mathrm{d}t + \sigma(Z_t, t)\mathrm{d}W_t) + \partial_t U(Z_t, t)\mathrm{d}t + \frac{1}{2}\mathrm{tr}(\nabla^2 U(Z_t, t)\sigma^2(Z_t, t))\mathrm{d}t,$$

for $t \in [0, T]$. Taking expectation on both sides,

$$\frac{\mathrm{d}}{\mathrm{d}t}\mathbb{E}(U(Z_t)) = \mathbb{E}[\nabla_z U(Z_t, t)^\top \eta(Z_t, t)] + \mathbb{E}\left[\partial_t U(Z_t, t) + \frac{1}{2}\mathrm{tr}(\nabla^2 U(Z_t, t)\sigma^2(Z_t, t))\right].$$

Let $u_t = \mathbb{E}[U(Z_t, t)]$. By the assumption above, we get

$$\dot{u}_t \leq -\alpha_t u_t + \beta_t + \gamma_t.$$

507 Following Grönwall's inequality (see Lemma [A.2](#) below), we have $\mathbb{E}[U(Z_1, 1)] = u_1 = \lim_{t \uparrow 1} u_t \leq$
508 $0$ if [(10)](#) holds. Because $U(z, 1) \geq 0$, this suggests that $U(Z_1, 1) = 0$ and hence $Z_1 \in A$ almost
509 surely. □

510 **Lemma A.2.** *Let $u_t \in \mathbb{R}$ and $\alpha_t, \beta_t \geq 0$, and $\frac{\mathrm{d}}{\mathrm{d}t}u_t \leq -\alpha_t u_t + \beta_t$, $t \in [0, T]$ for $T > 0$. We have*

$$u_t \leq \frac{1}{\zeta_t}(\zeta_0 u_0 + \int_0^t \zeta_s \beta_s \mathrm{d}s), \qquad \text{where} \qquad \zeta_t = \exp(\int_0^t \alpha_s \mathrm{d}s).$$

*Therefore, we have $\lim_{t \uparrow T} u_t \leq 0$ if*

$$\lim_{t \uparrow T} \zeta_t = +\infty, \quad \lim_{t \uparrow T} \frac{\zeta_t}{\int_0^t \zeta_s \beta_s \mathrm{d}s} = +\infty.$$

*Proof.* Let $v_t = \zeta_t u_t$, where $\zeta_t = \exp(\int_0^t \alpha_s \mathrm{d}s)$ so $\dot{\zeta}_t = \zeta_t \alpha_t$. Then

$$\frac{\mathrm{d}}{\mathrm{d}t}v_t = \dot{\zeta}_t u_t + \zeta_t \dot{u}_t \leq (\dot{\zeta}_t - \zeta_t \alpha_t)u_t + \zeta_t \beta_t = \zeta_t \beta_t.$$

So

$$v_t \leq v_0 + \beta \int_0^t \gamma_s \mathrm{d}s,$$

and hence

$$u_t \leq \frac{1}{\zeta_t}(\zeta_0 u_0 + \int_0^t \zeta_s \beta_s \mathrm{d}s).$$

To make $\lim_{t \uparrow T} u_t \leq 0$, we want

$$\lim_{t \uparrow T} \zeta_t = +\infty, \quad \lim_{t \uparrow T} \frac{\zeta_t}{\int_0^t \zeta_s \beta_s \mathrm{d}s} = +\infty.$$

511 □

**Corollary A.3.** *Let* $\mathrm{d}Z_t = \frac{x - Z_t}{1-t} + \varsigma_t \mathrm{d}W_t$ *with law* $\mathbb{Q}$. *This uses the drift term of Brownian bridge, but have a time-varying diffusion coefficient* $\varsigma_t \geq 0$. *Assume* $\sup_{t \in [0,T]} \varsigma_t < \infty$. *Then* $\mathbb{Q}(Z_1 = z) = 1$.

*Proof.* We verify the conditions in Theorem A.1. Define $U(z,t) = \|x - z\|^2 / 2$, and $\eta(z,t) = \frac{x - Z_t}{1-t}$. We have $\eta(z,t)^\top \nabla U(z,t) = -U(z,t)/(T-t)$. So $\alpha_t = 1/(T-t)$.

Also, $\partial_t U(z,t) + \frac{1}{2}\mathrm{tr}(\varsigma_t^2 \nabla_z^2 U(z,t)) = \frac{1}{2}\mathrm{diag}(\varsigma_t^2 I_{d \times d}) = \frac{d}{2}\varsigma_t^2 := \beta_t \leq C < \infty$.

Then $\zeta_t = \exp(\int_0^t \alpha_s \mathrm{d}s) = \frac{1}{1-t} \to +\infty$ as $t \uparrow T$.

Also, $\int_0^t \zeta_s \beta_s \mathrm{d}s \leq C \int_0^t \zeta_s \mathrm{d}s = CT(\log(T) - \log(T-t))$. So

$$\lim_{t \uparrow T} \frac{\zeta_t}{\int_0^t \zeta_s \beta_s \mathrm{d}s} \geq \lim_{t \uparrow T} \frac{\frac{1}{1-t}}{CT(\log(T) - \log(T-t))} = +\infty.$$

$\square$

Using Girsanov theorem, we show that introducing arbitrary non-singular changes (as defined below) on the drift and initialization of a process does not change its bridge conditions.

**Proposition A.4.** *Consider the following processes*

$$\mathbb{Q}: \quad Z_t = b_t(Z_t)\mathrm{d}t + \sigma_t(Z_t)\mathrm{d}W_t, \quad Z_0 \sim \mu_0$$
$$\tilde{\mathbb{Q}}: \quad Z_t = (b_t(Z_t) + \sigma_t(Z_t)f_t(Z_t))\mathrm{d}t + \sigma_t(Z_t)\mathrm{d}W_t, \quad Z_0 \sim \tilde{\mu}_0.$$

*Assume we have* $\mathcal{KL}(\mu_0 \| \tilde{\mu}_0) < +\infty$ *and* $\mathbb{E}_{\mathbb{Q}}[\int_0^T \|f_t(Z_{[0,t]})\|^2] < \infty$. *Then for any event A, we have* $\mathbb{Q}(Z \in A) = 1$ *if and only if* $\tilde{\mathbb{Q}}(Z \in A) = 1$.

*Proof.* Using Girsnaov theorem [31], we have

$$\mathcal{KL}(\mathbb{Q} \| \tilde{\mathbb{Q}}) = \mathcal{KL}(\mu_0 \| \tilde{\mu}_0) + \frac{1}{2}\mathbb{E}_{\mathbb{Q}}\left[\int_0^1 \|f_t(Z_t)\|_2^2 \,\mathrm{d}t\right].$$

Hence, we have $\mathcal{KL}(\mathbb{Q} \| \tilde{\mathbb{Q}}) < +\infty$. This implies that $\mathbb{Q}$ and $\tilde{\mathbb{Q}}$ has the same support. Hence $\mathbb{Q}(Z \in A) = 1$ iff $\tilde{\mathbb{Q}}(Z \in A) = 1$ for any measurable set $A$. $\square$

This gives an immediate proof of the following result that we use in the paper.

**Corollary A.5.** *Consider the following two processes:*

$$\mathbb{Q}^{x,\mathrm{bb}}: \qquad \mathrm{d}Z_t = \left(\sigma_t^2 \frac{x - Z_t}{\beta_1 - \beta_t}\right)\mathrm{d}t + \sigma_t \mathrm{d}W_t, \quad Z_0 \sim \mu_0,$$

$$\mathbb{Q}^{x,\mathrm{bb},f}: \qquad \mathrm{d}Z_t = \left(\sigma_t f_t(Z_t) + \sigma_t^2 \frac{x - Z_t}{\beta_1 - \beta_t}\right)\mathrm{d}t + \sigma_t \mathrm{d}W_t, \quad Z_0 \sim \mu_0.$$

*Assume* $\mathbb{Q}^{x,\mathrm{bb},f}[\|f_t(Z_t)\|^2] < +\infty$ *and* $\sigma_t > 0$ *for* $t \in [0, +\infty)$. *Then* $\mathbb{Q}^{x,\mathrm{bb},f}$ *is a bridge to* $x$.

# B  Model Details

## B.1  Model Architecture for Molecule Generation.

Following EGM [17], we apply an E(3) equivariant GNN network (EGNN) as our basic model architecture. EGNNs are a type of graph neural networks that satisfies the equivariance constraint,

$$\mathbf{R}x' + \mathbf{t}, h' = f(\mathbf{R}x + \mathbf{t}, h) \quad when \quad x', h' = f(x, h), \tag{11}$$

where $x$ and $h$ represent the 3D coordinates and additional features, orthogonal $\mathbf{R}$ stands for the random rotation and $\mathbf{t} \in \mathbb{R}^3$ is a random transformation. One EGNN is usually made up of multiple stacked equivariant graph convolutional layers (EGCL), and every EGCL satisfies the

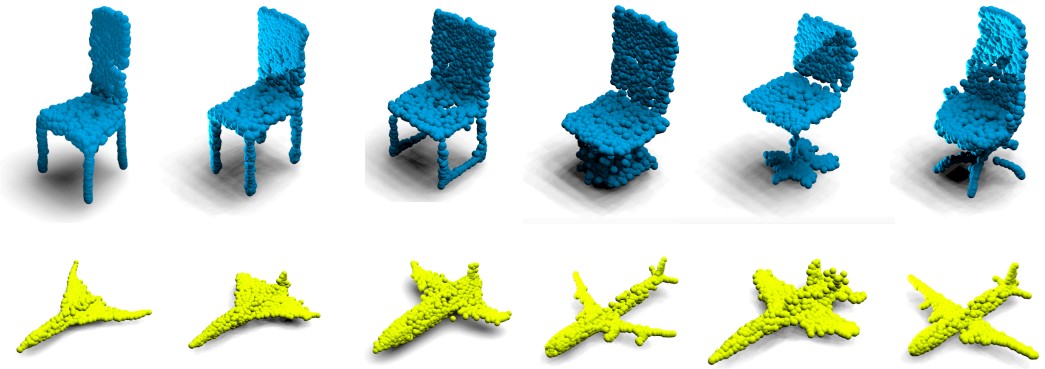

Figure 5: More visualization result of our Bridge-Statistic method, the upper row is chair category and the lower row is airplane category

equivariance constraint. Denote $N$ the number of nodes, $x^l$ and $h^l$ the coordinates and features for layer $l \in \{0, \cdots, L\}$, we have

$$
\begin{aligned}
m_{ij} &= \phi_e(h_i^l, h_j^l, d_{ij}), \\
h_i^{l+1} &= \phi_h(h_i^l, \{m_{ij}\}_{j=1}^N), \\
x_i^{l+1} &= x_i^l + \sum_{j \neq i} \frac{x_i^l - x_j^l}{d+1} \phi_x(h_i^l, h_j^l, d_{ij}),
\end{aligned}
\tag{12}
$$

where $h^0 = h, x^0 = x$, $d_{ij} = \|x_i^l - x_j^l\|_2$, $d_{ij} + 1$ is introduced to improve training stability, and $\phi_e, \phi_h, \phi_x$ represents fully connected neural network with learnable parameters. We refer the readers to the previous paper [34] for more details.

**Scaling Features**  Following [17], we re-scale the data with additional scaling factors. The atom type one-hot vector and atom charge value $\times.25$ and $\times 0.1$, respectively. It significantly improves performance over non-scaled inputs, e.g. $47\%$ relative improvements on molecule stability.

### B.2  Model Architecture for Point Cloud Generation.

We build up our network based on the setup in point cloud diffusion work [25] without extra modification for a fair comparison. The model consists two parts. The first part is a flow model that learns the shape prior and the second part takes the shape prior and the noisy point coordinates into a MLP style encoder as the denoise function. We refer the readers to the previous paper [25] for more details.

## C  More Visualization for Point Cloud Generation

Below we show more visualization of our point cloud generation result in both chair and airplane class. We focus on presenting our best performance Bridge-Statistic visualization in Figure 5.

## D  Discussion of Broader Impact

This research aims to generate molecules and point cloud samples with geometry prior guided bridge processes. It is possible to be beneficial for drug design, the food industry and many other fields. However, it might be used for generating harmful molecules and viruses.