# OpenReview forum: "Diffusion-based Molecule Generation with Informative Prior Bridges"
_NeurIPS.cc/2022/Conference — NeurIPS 2022 Accept_

### Official Review · Reviewer_cHTF · 2022-07-04

**Rating:** 7
**Confidence:** 4
**Soundness:** 3 good
**Presentation:** 3 good
**Contribution:** 4 excellent

**Summary:**

This paper introduces a novel diffusion-based generative model that leverages the prior information to learn the diffusion process satisfying the bridge condition. Specifically, the paper proposes a  Brownian bridge with extra drift term that incorporates the prior information, and aims to learn the diffusion process by fitting the trajectories drawn from the mixture of these bridges. The paper utilizes Lyapunov functions to guarantee the bridge condition, and further introduces energy functions to guide the training process for each downstream task. The proposed method, Bridge with Priors, is able to generate samples with better quality compared to the baselines in both molecule generation and point cloud generation tasks.

**Questions:**

1. Is Bridge with Priors designed for a more complex process such as Ornstein–Uhlenbeck process? Or is it applicable only to the Brownian bridge which is possible due to the singular characteristic of the drift term $\sigma^2_t\frac{x-Z_t}{\beta_1-\beta_t}$?

1. How to determine the learnable parameter $\alpha$ as in $s^{\theta}_t(z) = \alpha f_t(z)+\tilde{s}^{\theta}_t(z)$?

1. What is the meaning of "to ensure that $U$ is minimized at time $t=1$" in line 147?

1. Is the condition in line 157: $\mathbb{E}_{\mathbb{Q}^{x,bb}} [||f_t(Z_t)||^2] <\infty$  correct?
In the supplementary file, the condition of Proposition A.4. is $\mathbb{E} [\int^1_0||f_t(Z_t)||^2]  < \infty$.

1. Why is the molecular stability of GEOM-DRUG 0.0 for all models? Is it because the molecules are large?

1. Is the training time of Bridge with Priors larger than previous diffusion models? Additionally, although the comparison with the Schr&ouml;dinger Bridge methods [De Bortoli et al., 2021, Chen et al., 2022] is out-of-the-scope of this paper, I believe that the comparison would be interesting as Bridge with Priors may have less training/sampling time and competitive or better performance.

---
**References**
- De Bortoli et al., Diffusion Schrödinger Bridge with Applications to Score-Based Generative Modeling, NeurIPS 2021,
- Chen et al., Likelihood Training of Schrödinger Bridge using Forward-Backward SDEs Theory, ICLR 2022


**Limitations:**

Training the model takes a long time, which is a common limitation of diffusion models. The paper also discusses the future works in section 6, which seems to be a promising research direction.

**Strengths And Weaknesses:**

**Strength**

1. The paper proposes a novel approach of leveraging the prior information in the form of diffusion bridges to learn the diffusion process, which is clearly different from injecting inductive bias into the model architectures. Moreover, instead of using the time-reversal technique which most of the previous works are based on, the paper learns the generation process as a mixture of diffusion bridges which was introduced in [Peluchetti, 2022], which is also a new direction for the diffusion model.

1. The paper has a clear motivation of exploiting problem-dependent prior information to generate high-quality samples. Especially, for the point cloud generation, the paper addresses the problem of unevenly distributed points and provides a solution by using the Risze energy function.

1. Although the paper is based on mathematically heavy theory, the paper is well-written and understandable with clear notations and formulations, except for the part explaining the mixture of diffusion bridges and the loss function (lines 102-112).

1. The paper shows promising results for both molecule generation and point cloud generation tasks

1. I believe the experiments are well-conducted with clear data/training/evaluation setups and compared with the state-of-the-art baselines (EDM). Especially, comparing previous diffusion models, the Bridge without Prior, and the Bridge with Prior clearly shows the advantage of exploiting prior information for the generation.

**Weaknesses**

1. Lines 102-111 are too concise with necessary details omitted. Without reading [Peluchetti, 2022], it would have been impossible to understand the concept of "mixture of diffusion bridges" denoted as $\mathbb{Q}^{\Pi^{\ast}}$.

1. If I understood correctly, the term $b_t(Z_t|Z_1)$ in the loss function of Eq.(2) should be the drift term of the mixture $\mathbb{Q}^{\Pi^{\ast}}$, not the drift term of a specific $x$-bridge $\mathbb{Q}^{x}$. Is this term analytically accessible?

1. The computation of the loss function of equation (2) needs clarification. I assume the expectation in Eq.(2) was computed by a Monte Carlo estimate with a sample $(t,z_1,z_0,z_t)$ where $z_1\sim \text{Data}$, $x_0\sim\Pi_{0|1}(dx_0|x_1)$ and $z_t\sim p_{t|0,1}(z_t|z_0,z_1)$. Although [Peluchetti, 2022] shows that $p_{t|0,1}(z_t|z_0,z_1)$ has explicit form, how is $x_0\sim\Pi_{0|1}(dx_0|x_1)$ sampled?

1. The condition of line 157: $\mathbb{E}_{\mathbb{Q}^{x,bb}}\|f_t(Z_t)\|^2$ does not seem trivial which could be satisfied for most practical functions, although the provided intuition in lines 158-161 is convincing. It would be clearer if the authors provide proof that the condition is satisfied for $f$ induced from the proposed energy functions.

1. The reason for incorporating the energy function (prior information) into the Brownian bridge by $f_t(\cdot)=-\nabla E(\cdot)$ is not clear. Is the energy function minimized when $t$ approaches 1?

1. I would like to see more experimental results to the authors' claim for "less sampling time", either for GEOM-DRUG or the point cloud generation tasks. I can see that in Table 2, Bridge+Force with 500 steps outperforms EGM for 1000 steps in the QM9 dataset which is impressive, but as the molecules of QM9 are small, it would be more convincing if the claim is proven for larger datasets. It would be great if there is an ablation study section in the paper to emphasize the advantage of less sampling time.

(*) I would like to raise my score if the presentation is improved and the addressed concerns/questions are clarified.

**Minor correction**
- The explanation for the abbreviation SMLD is missing.
- The SDE of $\mathbb{P}^{\theta}$ and Eq.(1) use t-subscript notation for drift and diffusion coefficients, while Eq.(4) does not use such t-subscript. I recommend unifying the notation to prevent confusion.
- No bold in row Airplane, column 100 Steps-COV-EMD in Table 4.
- Condition #3 of Theorem A.1. in the supplementary file is missing.
- In my opinion, Algorithm 1 and Figure 1 do not give much information. It would be great to add more details about "mixture of diffusion bridges" and the explicit form of $b_t(Z_t|Z_1)$.

---
**References**
- Peluchetti, Non-denoising forward-time diffusions, 2022

---

> ### Author Response · Authors · 2022-08-02
> **Response to Reviewer cHTF [Part 1]**
>
> Thanks for giving us careful comments and suggestions. We provide pointwise responses to your concerns and questions about our paper below:
>
> **Weakness 1.** Lines 102-111 are too concise with necessary details omitted
>
>
> **A:** We tried to give a concise description of the main idea of bridge models here, which is difficult due to the advanced stochastic calculus tools involved. We will add a more thorough introduction to the appendix in the final version.
> Essentially, given a set of bridges $\mathbb{Q}^x$ that guarantees to have $Z_1  = x$ for each $x$, we want to train the neural process to approximate the mixture of $\mathbb{Q}^x$ when $x$ is drawn from the data distribution. In this way, the $Z_1$ generated from the neural process would also follow the data distribution.
>
> **Weakness 2.** the term $b_t(Z_t|Z_1)$ in the loss function of Eq.(2) should be the drift term of the mixture $\mathbb{Q}^{\pi*}$.  Is this term analytically accessible]
>
> **A:** It is correct that, theoretically, we should have $s_t^\theta(Z_t)$ to match $b_{t(Z_t)}^{\Pi^*}$, the
> drift of $\mathbb{Q}^{\Pi^*}$. But we can show that  $b^{\Pi^*}(Z_t) = \mathbb{E}_{Z_1 \sim \Pi^*}[b_t(Z_t|Z_1)]$, and hence (ignoring the variance term for simplicity):
>
> $$
> \mathbb{E}\left[ || {s_t^{\theta(Z_t)} - b_t^{\Pi^*}(Z_t|Z_1)} ||_2^2 \right] = \mathbb{E}\left[||{s_t^{\theta(Z_t)} - b_t^{\Pi^*}(Z_t|Z_1)}||_2^2 \right]  + const.
> $$
>
> Therefore, it is equivalent to match $s_t^\theta(Z_t)$ with  $b_t^{\Pi^*}(Z_t|Z_1)$.
> The identity above is due to
> $$
> \mathbb{E}[||{X^\theta-Y}||^2] =
> \mathbb{E}[||t{X^\theta-\mathbb{E}[Y]}||^2] + \mathrm{Y},
> $$
> and $\mathrm{Y}$ is independent with $\theta$.
>
>
> **Weakness 3.** The computation of the loss function of equation (2) needs clarification
>
> **A:** You are correct regarding the evaluation of Eq (2),
> except that $x_0$ can be drawn from **any initial distribution** (e.g., standard Gaussian), once $\mathbb{Q}^x$ is a bridge process that converges to $x$ regardless of the initialization.  We will clarify this part.
>
>
> **Weakness 4.** The condition of line 157 does not seem trivial which could be satisfied for most practical functions
>
> **A:** $\mathbb{E}[\left\lVert{f(Z_t)}\right\lVert_2^2] < +\infty$ (regarding any distribution of $Z_t$) is trivially satisfied if $f$ is bounded, i.e., $\sup_z \left\lVert{f(z)} \right\lVert<+\infty$.
> It can also be easily satisfied for unbounded $f$ if $Z_t$ has bounded moments, e.g., when $\left\lVert{f(x)}\right\lVert\leq C\left\lVert{x}\right\lVert^\alpha$ for some $C<+\infty$ and $\alpha\in \mathbb R$ (true for  ReLU networks with $\alpha = 1$)  and $\mathbb{E}[\left\lVert{Z_t}\right\lVert^\alpha] <+\infty$.
>
>
> **Weakness 5.** Is the energy function minimized when t approaches 1
>
> **A:** The physical energy term is **NOT** minimized at $t=1$, because we would have $Z_1 = x$ guaranteed when following the $x$-bridge $\mathbb{Q}^x$ (instead, $Z_t = x$ minimizes the Lyapunov function, which is the sum of the energy term and a singular term $\frac{Z_t-x}{\beta_1-\beta_t}$).
> The goal of incorporating the physical energy term is to regularize the trajectory of $Z_t$ before it hits $t=1$,
> so that the neural generative process learns from $\mathbb{Q}^x$ also has more ``physical looking" trajectories.
> It is an empirical finding that much more physically regularized processes yield better learning performance.
>
>
> **Weakness 6.** More experimental results to the authors' claim for "less sampling time"
>
> **A:** 1) The advantage of the faster sampling speed of our method is also significant, as shown in Table 4:
> Both the MMD and COV score of our method with 10 steps matches or even better when compared with the Diffusion or Bridge baseline model with 100 steps (e.g., our 10-step Chair COV-CD is even better than the 100-step Diffusion baseline and our 10-step Chair MMD-EMD is better than 100 step Bridge baseline ).
> 2) We additionally list the few-step result on GEOM-DRUG. We notice that, 200-step Bridge + Force achieves comparable results as 1000-step E-GDM.
> 3) We will properly add an ablation study section for the less sampling step and the sampling speed (See response for reviewer nq85 for the current sampling speed comparison)  in our next revision.
>
> | Method / Atom Stable  | 200 Step | 1000 Step |
> |:-|:-:|:-:|
> | Bridge + Force| 0.812 | 0.824 |
> | E-GDM | 0.798 | 0.813 |

---

> > ### Author Response · Authors · 2022-08-02
> > **Response to Reviewer cHTF [Part 2]**
> >
> > **Question 1.** Is Bridge with Priors designed for a more complex process such as Ornstein–Uhlenbeck process
> >
> > **A:** An $x$-bridge is any process $Z$ that guarantees to achieve $Z_1 = x$ at $t=1$. There are different ways to construct processes that satisfy such conditions. A typical approach to constructing such processes is to take an arbitrary process, denoted by $\mathbb M$, such as an Ornstein-Uhlenbeck process, and derive its conditional process $\mathbb M(\cdot |Z_1 = x)$ when its endpoint is pinned at $x$; existing methods, such DDPM, SMLD, and the method in Peluchetti can be viewed in this way.
> > In brief, the conditioned OU processes can be used as a $x$-bridge.
> >
> > However, the conditioning method requires mathematical derivation and is restricted to simple processes with a closed form. The main contribution of this work is to show that, by developing a more general Lyapunov criterion, we can construct much more flexible bridges that incorporate complex physical prior information.
> >
> > We will try our best to clarify these points in the revision. We hope the reviewer can understand that it is a challenging task given the limited space and the application-oriented scope of the paper.
> >
> >
> > **Question 2.** How to determine the learnable alpha
> >
> > **A:** $\alpha$ is trained together with $\theta$ to minimize the loss in a typical way.
> >
> >
> > **Question 3.** What is the meaning of to ensure that U is minimized at time t = 1 in line 147]
> >
> > **A:** Here $U$ is a Lyapunov function that we use to certify $Z_t = x$. Hence, we want that $U(\cdot, t=1)$ to be minimized at $x$ by construction.
> >
> > **Question 4.** Is the condition in line 157 correct?
> >
> > **A:** It is correct and it is consistent with Corollary A.5; Proposition A.4. is used a Lemma. We will clarify this.
> >
> > **Question 5.** Why is the molecular stability of GEOM-DRUG 0.0 for all models?
> >
> > **A:** The stability is a difficult problem for all existing methods, which is an open question that we hope to address in future works. Essentially, the stability checks if the molecules satisfy the union of a set of constraints (e.g., the distances of a bound are in a certain region), and it reports failure once one constraint is not met. In future works, we will investigate how to incorporate the hard stability constraints as priors in generative models.
> >
> >
> > **Question 6.** Is the training time of Bridge with Priors larger than previous diffusion models & compare with Schrödinger Bridge
> >
> > **A:** Priors larger than previous diffusion models;  compare with Schrödinger Bridge]
> > We conduct all our experiments with the same training epoch as the baseline. In addition, as we reply to Reviewer nq85 in Question 1, computing the energy term only yields a minor extra time cost (around $3\%$, $1.18s$ vs. $1.22s$). Hence, in practice, our bridge's inference and training time with priors are almost the same as the previous diffusion models.
> >
> > Schrödinger Bridges (SB) is an alternative approach to diffusion generative models. However, the training process of SB is more complicated and expensive since it requires solving an entropy regularized optimal transport problem, while our method only specifies an arbitrary bridge process.  Importantly, we leverage the flexibility of bridge processes to incorporate prior information, but it is unclear how to do this in SB. We will add a discussion regarding this issue.

---

> > > ### Comment · Reviewer_cHTF · 2022-08-05
> > > **Thanks for the Response**
> > >
> > > Thank you for the detailed response and clarification.
> > >
> > > My previous concerns regarding the loss function have been fully addressed and I would like to raise my score from 6 to 7.
> > > The paper provides a novel framework for injecting prior information into the diffusion process, which is flexible for diverse domains and prior information. The authors have shown two successful applications with improved results and less sampling time, which also seem promising for other different tasks. Thereby, I would like to raise my score further if the authors provide a revised paper with the mentioned clarifications.
> > >
> > > One last question:
> > > From what I have understood, the proposed Lyapunov function method (in section 3.2) provides a criterion for a general form of bridge processes. Is the Lyapunov function method related to the injection of prior information? If so, It would be great if the authors provide some intuition about the relation between the Lyapunov function and the injection of prior knowledge, especially in the form of physical energy.

---

> > > > ### Author Response · Authors · 2022-08-09
> > > > **Thanks for raisng the score**
> > > >
> > > > Thanks for raising the score and giving positive feedback on our work. We submit a revised version and trying our best to cover as much as clarities as possible in blue in this version. Since the page limit is still nine pages at the current stage, we are running out of space to cover all the clarification above. We will fully cover them if this paper can get accepted with one additional page.
> > > >
> > > > For your question, The Lyapunov function method is a technique to determine and verify whether the bridge condition will hold when we add a force term on the Brownian bridge for injecting prior knowledge.

---

### Official Review · Reviewer_3WV1 · 2022-07-09

**Rating:** 6
**Confidence:** 2
**Soundness:** 2 fair
**Presentation:** 2 fair
**Contribution:** 2 fair

**Summary:**

This paper proposes a framework for incorporating physics driven prior bridges in diffusion models for improved molecule and point cloud generation. In order to do so, a Lyapunov function based method is developed to construct and determine bridges. The work seems to improve on existing methods for molecule and point cloud generation.

**Questions:**

See questions in weaknesses sections above.

**Limitations:**

The limitations are mentioned in the weaknesses section above.

**Strengths And Weaknesses:**

***Strengths***
1) The proposed contribution of Lyapunov function to construct prior bridges is novel in my opinion and is a sensible approach for incorporating informative priors.

2) The proposed method seems to improve molecule generation and point cloud generation benchmarks.

***Weaknesses***

1) For GeomDrug datasets, although the atom stability seems to improve, the overall molecule stability seems to be 0%, i.e., none of the predicted molecules are correct. Although this is the case with other existing models as well, it is surprising to see that the proposed method using informative physics informed priors does not improve this aspect of generation. Will inferring the edges during the diffusion process (instead of doing so in a post processing manner with RDKit) improve this aspect?

2) I did not find any mention of computational complexity of the proposed method. It would be nice to see how feasible is the proposed approach from a practical perspective. In drug discovery, often a deserted outcome is the de novo generation of molecules at a reasonable speed. Hence, knowing about the computational requirements and the time taken on average for generating realistic molecules will be a good information to have.

---

> ### Author Response · Authors · 2022-08-02
> **Response to Reviewer 3WV1**
>
> Thanks for giving us careful comments and suggestions. We provide pointwise responses to your concerns and questions about our paper below:
>
> **Weakness1.** the overall molecule stability seems to be 0%
>
> **A:** The stability is a difficult problem for all existing methods, which is an open question that we hope to address in future works.  Essentially, the stability checks if the molecules satisfy the union of a set of constraints (e.g., the distances of a bound are in a certain region),
> and it reports failure once one constraint is not met.  We also mention this as a limitation and future direction in line 327-328.
>
> Moreover, our prior is imposed on the trajectory that generates the molecules in the diffusion process, and hence its impact on the constraint satisfaction of each bond could be small. We think that we would need to explicitly incorporate the hard constraints in the stability evaluation to improve the stability.
>
>
>
>
> **Weakness2.** Computational complexity of the proposed method
>
> **A:** The per-step computational cost of our method is almost the same as that of standard diffusion models during inference. This is because our method and the baselines train the same neural diffusion models except for the extra energy term (a simple function involving no neural networks and is fixed during training). Hence, our method yields a lower total inference time, giving comparable results with fewer inference steps.

---

> > ### Comment · Reviewer_3WV1 · 2022-08-08
> > **Thank you for your response**
> >
> > I thank the authors for their responses to my questions. I will keep my score the same since I have a relatively low confidence on my assessment compared to the other reviewers. However, I would acknowledge that I find the paper addresses a useful problem and in my opinion if the overall molecular stability (which is at 0% currently and has been slated for future work) can be improved, then it can have a strong impact in drug discovery domain. So I would maintain my weak support for the acceptance of the paper.

---

### Official Review · Reviewer_nq85 · 2022-07-10

**Rating:** 6
**Confidence:** 4
**Soundness:** 3 good
**Presentation:** 3 good
**Contribution:** 3 good

**Summary:**

This paper proposes a new diffusion method incorporating physical information by carefully designing the prior for the downstream task. The suggested diffusion approach incorporates physical prior into bridges, in contrast to other diffusion methods that learned diffusions as a combination of forward-time diffusion bridges. The method was evaluated on molecules and 3D point cloud generation tasks, results of which show advantages in terms of generating quality, efficiency, and energy calculation convenience. They demonstrate the superiority of the method in terms of both quantity and quality.


**Questions:**

* Taking fewer time steps sometimes could mislead because actual wall clock time spent for each time step could be different. Wondering if a wall clock comparison can be presented.


**Limitations:**

* The scientific significance of merely producing molecular conformation is limited. The usefulness of this work would be higher if this model could be applied to real-world drug discovery applications, as the authors emphasized.

**Strengths And Weaknesses:**

### Strength
* The paper is generally clear, well structured, and easy to follow.
* The theory and accompanying research are explained clearly.
* They properly motivate the need for the method and describe the proposed model thoroughly.
* Injecting task-dependent priors into a diffusion method is non-trivial and novel
* They reasonably select the benchmarks emphasizing the strength of the suggested model.


### Weakness
* Typos
  * Line 42: repeated citation [14]
  * Line 109: ofdenoised -> of denoised
  * Line 196: bound(x) -> bond(x)
  * Line 240: [26] seems to be an incorrect citation because it's nothing to do with a molecule application.
* Table 1 omits uniqueness comparisons that are important to determine the superiority of the model. Including uniqueness comparisons is important because there may be models that generate molecules with low uniqueness and high novelty only. (I'd be happy to increase my score if this issue is resolved)

---

> ### Author Response · Authors · 2022-08-02
> **Response to Reviewer nq85**
>
> We thank the reviewer for your time and comments. We address all your concerns as below:
>
> **Weakness 1.** Some Typos
>
> **A:** Thanks for pointing out the typos. We will carefully proofread the draft in the revision.
>
> **Weakness 2.** Table 1 omits uniqueness comparisons
>
> **A:** Here, we show the uniqueness metrics in Table 1, which is the percentage of valid and unique molecules in 12000 generated molecules. We see that our method outperforms E-GDM and EN-Flow. We copy the EN-Flow results from [17] because we do not run an EN-Flow model.
> | Method | Valid + Unique |
> |:-|:-:|
> | EN-Flow| 0.349|
> |E-GDM |0.902|
> | Bridge |0.902 |
> | Bridge + Force | 0.907 |
>
>
> **Question 1.** Taking fewer time steps sometimes could mislead & if a wall clock comparison can be presented.
>
> **A:** Because our method and the baselines train the same neural diffusion models except the extra energy term (which is a simple function that involves no neural networks and is fixed during training), the wall clock time is very close for our method and the baselines. Hence, the number of steps can be viewed as a good surrogate of wall clock time.
>
> The table below shows the time spent by our method and E-DGM on each step (difference <3%). The slight increase in time of the method is due to the need to load the prior function. We also list the per-epoch training time difference when we use AMBER prior. We use 9% more time than the baseline because of the calculation of prior force.
>  We will add this table to the paper in our next revision.
>
> | Method | One-step Inference Time (Second) | One-batch Training Time (Second) |
> |:-|:-:|:-:|
> | E-DGM | 1.18 | 98.1 |
> | Bridge| 1.22 | 107.3 |
>
>
>
> **About the Limitation:**
> Since the main goal of this work is to propose a novel methodology, we choose molecule conformation and point cloud generation as two examples of areas where our method can contribute to improvement. It is a part of our plan to apply the method to solve the more challenging drug design problems requiring collaboration with domain experts.

---

> > ### Comment · Reviewer_nq85 · 2022-08-09
> > **Response to Rebuttal**
> >
> > I appreciate your thorough response and explanation.
> >
> > Since all of my prior issues have been resolved, I would like to increase my rating from 5 to 6.

---

### Meta-Review · Area_Chair_Ffpp · 2022-08-26

**Recommendation:** Accept
**Confidence:** Certain

**Metareview:**


All reviewers agreed that this work has many positive aspects, such as originality of the idea, technical soundness and practical relevance. In the initial reviews, some concerns about the experimental evaluation have been raised. In particular, one reviewer mentioned potential problems regarding the uniqueness of generated molecules. This issue, however could be addressed reasonably well in the rebuttal.
I do share the generally positive perception of this paper. Therefore, I recommend to accept to paper.

**Award:**

No

---

### Decision · Program_Chairs · 2022-09-14

Accept